# Antecedents of Psychological Well-Being among Swedish Audit Firm Employees

**DOI:** 10.3390/ijerph17103346

**Published:** 2020-05-12

**Authors:** Pernilla Broberg, Torbjörn Tagesson, Timur Uman

**Affiliations:** 1Department of Management and Engineering, Business Administration Division, Linköping University, 58183 Linköping, Sweden; torbjorn.tagesson@liu.se; 2Jönköping International Business School, Jönköping University, 55111 Jönköping, Sweden; timur.uman@ju.se; 3Department of Business, Kristianstad University, 29188 Kristianstad, Sweden

**Keywords:** psychological well-being, audit firm employees, auditors, business professionals, GHQ-12

## Abstract

The attractiveness of audit firms as employers appears to have decreased in recent years and the audit profession is currently experiencing high employee turnover. A shortage of personnel increases the risk of long-term stress and illness. This paper therefore proposes audit firm employees’ well-being as an important research topic and explores the antecedents of well-being of Swedish audit firm employees in comparison with those of other business professionals. Based on a nationwide survey of members of the Swedish association of business professionals, with a focus on psychological well-being (measured through General Health Questionnaire-12 (GHQ-12)), the study shows that the psychological well-being of the professionals in this study generally aligns with the results from similar studies in a Swedish context. However, the findings indicate that audit industry respondents have the lowest psychological well-being and that employer change, job satisfaction, and life satisfaction were the strongest antecedents of their psychological well-being.

## 1. Introduction

Although students rank audit firms highly when ranking future and potential employers in surveys, the attractiveness of audit firms appears to have decreased in recent years [1]. Furthermore, the Swedish audit industry is characterized by high employee turnover [2,3,4], and of those who are annually employed in this industry, less than a third stay long enough to take the auditor exam [5]. According to the Swedish Inspectorate of Auditors, the number of certified auditors has continually decreased since the late 1990s. People choose to leave this industry and change their careers for different reasons: (i) natural turnover among first-step career climbers, where employees enter the industry as a first step in their careers, but have no intention to stay [6]; (ii) turnover due to the “up-or-out” system—i.e., employees are expected to make a career and climb the audit firm’s career ladder and the most talented and best-suited employees will have the opportunity to climb to higher positions within the organization, while less competent and productive employees are expected to leave [7,8,9,10]; (iii) work-related dissatisfaction regarding desires, expectations, and demands, such as a lack of job satisfaction [11,12,13]; and (iv) competition on the labor market for accountants and business administrators. All of these reasons for employee turnover are likely to be strengthened in times when there is a high demand for labor. A shortage of personnel increases the risk of long-term stress and illness [14,15,16] and could be another reason why people may leave the labor market or change careers. Thus, in line with previous research, we assume that the relationship between job satisfaction and employee turnover is affected and moderated by job availability [17]. In addition, it could be assumed that turnover and personnel shortage affect workers’ ability to maintain a satisfactory level of performance [18,19]. In turn, this adversely affects job satisfaction in particular, and well-being in general [16,20,21,22]. 

While a number of studies have explored the antecedents of the well-being of business professionals, and of auditors in particular, these studies have primarily reported a satisfactory level of well-being among auditors [23,24]—a finding that contradicts the reality described in the media and in interviews with auditors. Reasons for this discrepancy might include: (i) a self-selection bias in small samples used previously; and (ii) salutogenic-inspired well-being measures such as job satisfaction and life satisfaction that do not take into account the ill-being dimension in individuals’ subjective appreciation of their well-being. Addressing the discrepancy between empirical reality and previous research, the aim of this study is to explore the psychological well-being of Swedish audit firm employees and compare it with that of other business professionals. This is done through a nationwide survey of members of a Swedish trade union of business professionals, thus reducing potential self-selection bias as well as using multiple instruments to measure well-being that are both salutogenic and non-salutogenic in nature. Furthermore, instead of studying audit firm employees as an isolated professional group, this study explores the antecedents of this group’s well-being in comparison with those of other groups within the widely defined business profession. In exploring the interrelation between different triggers of psychological well-being we rely on the seminal study by Danna and Griffin [25] that doesn’t only distinguish between different facets of well-being in an organizational context and their interrelations but also suggest how demographic and context-specific characteristics are associated with these facets. The audit profession is characterized by constant change and a high degree of pressure on individual auditors [26]. In the following sections, several such changes and causes of pressure are described, along with some potential consequences of their presence in the audit profession.

Due to regulations such as the EU reform package, including Directive 2014/56/EU (amending Directive 2006/43/EC) and Regulation (EU) No 537/2014 [27], the audit industry has been and is still undergoing change. Auditors and audit firms are facing new requirements, prohibitions, limitations, and increased control aimed at improving the quality of statutory audits (these include a variety of issues, such as new audit reports, auditor/firm rotation, and non-audit services). At the same time, the audit profession in Sweden, as well as in other countries, is experiencing deregulation through the abolition of the statutory audit for many companies and significantly reduced education requirements for future auditors. 

Changing market conditions due to factors such as global competition and industry deregulation are affecting the audit profession, along with market, customer and business process orientation, and the increasing commercialization of audit services [28,29,30,31].

Upcoming challenges to the profession are presented by advances in information technology (IT), digitalization, and automation in terms of machine learning and artificial intelligence (AI) [32]. These phenomena are likely to cause major changes in auditor work; for example, auditor judgment, which has traditionally been seen as a crucial part of professional work, may be replaced by qualified IT systems encompassing big data and machine learning [33,34]. 

The audit profession is, as explained in the above, characterized by an “up-or-out” system [7,8,9,10] and also by a reward structure founded on the basic ideas of tournament theory [35]. To a certain extent, the audit profession involves “natural” turnover; however, its use of tournament theory seems to be one of the reasons behind a tough working environment that reduces job satisfaction and causes (too) many junior auditors to leave the profession [2]. Such an employee turnover can result in reduced expertise and is thus a threat to audit quality [12,36]. Factors behind turnover and reduced job satisfaction include an increased emphasis on business activities at the expense of professional activities and values [2]. 

Several studies have shown that time budget pressure (TBP) characterizes and influences auditors’ work and work environment [37,38,39,40,41]. TBP influences individual auditors’ efficiency, productivity, and personal health and thus causes negative stress due to a feeling of not meeting expected performance requirements [42,43]. It also causes reduced audit quality due to quality-threatening/-reducing behaviors such as accepting weak client explanations and premature signoffs [44]. 

Furthermore, audit quality is a rather vague “goal” that is difficult to gauge [9]. It has been argued that auditor reputation and trustworthiness are the only ways to guarantee the quality of an audit [38,39,40]. In the aftermath of major accounting/auditing/business scandals, the pressure on auditors and audit firms is therefore especially high. 

The examples above show that the audit profession is characterized by the simultaneous presence of different competing and/or complementing focuses and orientations. The professional orientation is challenged by, for example, market, customer, and business process orientations. This means that professional values such as auditor independence [45,46] and business values such as customer satisfaction [28,29,30,47] are simultaneously affecting auditors in their daily work. Previous research has shown that auditors are involved not only in “traditional” audit work but also in what can be described as marketing activities [48]. Studies have also found the simultaneous presence of professional and organizational identity among auditors [31]. 

Several of the issues and aspects described above are also present in other industries and other professions [49]. Yet it is likely that the audit industry, and auditors, are particularly exposed, mainly due to the contradictory nature of several changes and expectations that are present at the same time.

## 2. Method

### 2.1. Data Collection and Participants

The design of this study involved a quantitative cross-sectional approach. Data collection was carried out from October to December 2017 in cooperation with Civilekonomerna, a Swedish trade union of business professionals with approximately 43,120 members (mainly business professionals), which annually surveys its members’ wages and employment situation. The survey was administered and distributed to approximately 31,000 members via e-mail by the trade union. In collaboration with staff responsible for the annual survey at Civilekonomerna (mainly one investigator and one statistician), we were able to add and design a specific section of questions measuring well-being (which were previously not included in Civilekonomerna’s annual survey). The data from these questions provides the dependent variable of this study. In addition to this data, we had access to all other data from the survey. However, as the questions and measurements were decided by Civilekonomerna, according to their annual survey form, we had no influence on those questions and variables. The independent variables of this study are thus decided by availability and measurements used in the Civilekonomerna survey data set. Yet, in selecting the variables for this study we were guided by a seminal study by Danna and Griffin [25] that has outlined a framework for studies of well-being in an organizational context. The survey resulted in approximately 11,000 respondents in total, of which 6676 were useful for this study. That is, these respondents answered the questions regarding well-being and were working full- or part-time when the survey was distributed, while respondents who were unemployed during this time period were excluded. The number of useful respondents was reduced also due to the fact that a relatively large number of respondents did not answer questions regarding industry (3827 respondents), age (1350 respondents), gender (1349 respondents), and approximately 1000–1100 non-responses on the remaining independent variables.

### 2.2. Measurements 

To measure the dependent variable—namely, psychological well-being—the General Health Questionnaire (GHQ-12) was used. GHQ-12 is a widely-used unidimensional self-assessment instrument that focuses on the last two weeks and uses 12 questions to measure psychological health problems including the dimensions of (i) anxiety and depression, (ii) social dysfunction, and (iii) loss of confidence [50]. GHQ-12 includes questions regarding the ability to concentrate, loss of sleep (due to worry), playing a useful part in things, being capable of making decisions, being constantly under strain, facing up to everyday problems, being reasonably happy, being unable to overcome difficulties, being unhappy or depressed, losing confidence in oneself, thinking of oneself as worthless, and enjoying day-to-day activities. The respondent is asked to answer each of the 12 questions using a four-point scale (0 = not at all, 1 = the same as usual, 2 = rather more than usual or 3 = much more than usual). 

The GHQ-12 score can be calculated in different ways. This study used the Likert method, in which the GHQ-12 score can vary from 0 to 36 points, which provides a broad range of points that can be used when measuring the degree of psychological health and when comparing groups. A high score indicates a potential risk for psychiatric problems, such as low psychological well-being, in this study.

To interpret GHQ-12, normal values and limit values can be used. These values vary slightly depending on the area of use. Normal values are based on the average (and standard deviation) of a larger group (representing a particular population), and limit values are obtained by comparing groups with and without identified problems. Normal values and limit values should generally be used with caution. The scores may vary according to population and according to previous studies [51]. GHQ-12 has been shown to be sensitive to gender, social class membership, and whether the respondent is working or unemployed. Cultural differences in respondents can also affect the measure to a certain extent. The data set used in the study does not include information that enables comparison between groups with and without problems, therefore only normal values are used to interpret GHQ-12.

This study also included two additional measures of well-being, job satisfaction, and life satisfaction, which have been highlighted as important in terms of professional well-being in an organizational context, as they capture work-specific dimensions and dimensions of life as a whole, respectively [52,53]. These variables were measured with a single question each (i.e., How satisfied or dissatisfied are you with your current job/life?) and a seven-point Likert scale (where 1 = not at all satisfied and 7 = completely satisfied) was used for the response options. 

To explore the antecedents of psychological well-being, this study used both demographic variables and employment-related variables (that had previously been found to affect health and well-being). The demographic variables include gender (female/male), age, and monthly salary (self-reported actual amount in TSEK per month before tax). The employment-related variables include employment change (0 = the same employer as one year ago, i.e., October 2016, 1 = changed employer since October 2016), overtime compensation (0 = no overtime compensation, 1 = overtime compensation in the form of money and/or time off), overtime hours (0 = never works overtime, 1 = 1–8 h per month, 2 = 9–16 h per month, 3 = 17–31 h per month and 4 = more than 32 h per month), manager (0 = not manager, 1 = operational, budget and/or personnel manager) and industry (audit, service, staffing, IT, telecom, bank, insurance, trade, manufacturing, non-profit and other). 

### 2.3. Statistical Analysis

To explore the data, IBM SPSS Statistics 25 software (IBM Corporation, New York, NY, USA) was used. Descriptive statistics, independent sample t-tests, and Pearson chi-square tests (to compare subgroups), Pearson correlation and multiple linear regression analyses were employed.

## 3. Results

Table 1 provides the descriptive statistics for the whole sample (*n* = 6676), for the group excluding respondents from the audit industry (*n* = 6228) and for respondents from the audit industry (*n* = 448). The column to the far right shows differences between the subgroup of respondents from the audit industry and the subgroup consisting of the other respondents. The mean GHQ-12 score for the whole sample is 9.66. For the subgroup of respondents from the audit industry, the GHQ-12 score was higher, at 10.43. 

For the whole sample, the average age is 42.7 years with a standard deviation of 10. Slightly more women than men are included in the sample. Just over one-fifth of the respondents have changed employer during the last year. More than half of the respondents have no right to overtime compensation, even though less than a third have a managerial position. The average monthly salary amounts to 55,000 SEK, with a standard deviation of 26. The mean scores for job satisfaction and life satisfaction were M = 4.97 and M = 5.56 respectively.

The comparison between the two subgroups indicates that the respondents from the audit industry differ significantly in many aspects from the other respondents in the sample with one exception being job satisfaction where the differences are only weakly significant. 

Table 2 indicates significant correlations between the dependent variable and a majority of the independent variables. The correlation matrix indicates no multicollinearity problems, as none of the pairwise correlations are close to the critical value of 0.8 [54]. 

Table 3 presents the regression models. Regression Model 1 includes the whole sample, while Model 2 excludes the respondents from the audit industry. Model 3 includes only the respondents from the audit industry. 

Model 1 shows that the dependent variable, GHQ-12, is significant and is positively associates with the variables gender and overtime. The variables age, overtime compensation, salary and manager are significant and are negatively associated with the dependent variable, GHQ-12. None of the dummy variables for industry are significant in this model. 

Aside from the fact that the variable *age* is not significant in Model 2, the results for Model 2 are in line with the results for Model 1. However, in Model 3, which only includes respondents from the audit industry, the results are somewhat different. Unlike the other models, the variables gender, overtime compensation, overtime, and manager are not significant or associated with the dependent variable, GHQ-12. However, for the respondents from the audit industry, the variable employer change is significant and is negatively associated with the dependent variable, in line with Models 1 and 2, the variables salary, job satisfaction and life satisfaction are significant and are negatively associated with the dependent variable, GHQ-12. In comparison with the other models, these variables have an even stronger explanatory power in Model 3. 

All three reported models were significant at the 0.001 level. The models’ explanatory power, R^2^, varied between 40.5% and 44.5%.

Separate regression analyze were conducted for the other industries. The results of these models showed that job satisfaction and life satisfaction have an impact on psychological well-being in all 11 industries. 

To test for the robustness of the regression models several regressions analyses, using different combinations of the independent variables, were performed. A regression model including audit industry as the only industry variable (i.e., audit industry = 1, all other industries = 0) showed the same result as the models presented in Table 3. Further robustness tests were performed by (i) excluding all industry variables, (ii) excluding each independent variable separately in the regression analyses. These tests were performed to exclude the possibility that the well-being results are explained by other subgroups or characteristics and in order to able to take into account the possible effect of different numbers of non-responses for different variables. All the regression models passed the robustness tests.

## 4. Discussion

The aim of this study was to explore the psychological well-being of Swedish audit firm employees compared with that of other business professionals. The results of the study indicate that audit firm employees have a lower psychological well-being than business professionals in general. The results also indicate that respondents from the audit industry are significantly younger, worked more overtime, were more prone to changing employers, had lower salary, and were less likely to be in a managerial position than respondents from other business professions. Furthermore, the results show that audit firm employees score their job satisfaction to be high and were similar in their scores to the other business professionals, yet they score their life satisfaction lower than the other business professionals. 

Results of the multivariate analysis for all business professionals show that female respondents have lower psychological well-being than men, and overtime negatively affects psychological well-being. The results also suggest that older respondents have a higher level of psychological well-being than younger respondents, that higher pay and overtime compensation have a positive impact on psychological well-being, and managers experience higher levels of psychological well-being than other respondents. Finally, the study shows that job satisfaction and life satisfaction are the strongest predictors of psychological well-being of business professionals. Results of the multivariate analysis on audit firm employees as a group revealed that aspects such as gender, overtime compensation, overtime, or being a manager are not associated with this group’s psychological well-being. Compared to other business professionals, the audit firm employees’ well-being is instead triggered by employer change, indicating that respondents who have changed employers during the last year have a higher level of psychological well-being than those who remain with the same employer. Finally, the results suggest that salary, job satisfaction as well as life satisfaction are stronger predictors of audit firm employees’ psychological well-being compared to the other business professionals. 

The findings of our study related to job and life satisfaction of auditors resonate well with previous research, suggesting that auditors have a relatively high average level of both [23,24]. Our findings can also be compared to the findings within the financial services industry [55]. The results show that job and life satisfaction are positively associated with psychological well-being and are in line with previous research in other settings [56].

The indication of lower psychological well-being among auditors compared to other business professionals, might be one of the reasons why the audit profession is faced with higher turnover rates [2,4]. This interpretation is also supported by our findings that auditors’ psychological well-being is positively associated with the change of the employer. These findings might also reflect the unique career ladder manifested in the up-or-out system [7,8], which might lead to decreased psychological well-being due to the dissonance between one’s effort and the perceived fairness of remuneration for this effort and career advancements. The same argument, embedded in the industry career advancement structures [7,8], might be applicable to the findings that, neither overtime compensation, nor overtime or management position are associated with psychological well-being of auditors. However, these factors are associated with psychological well-being of other business professionals. Finally, indication that salary triggers auditors’ psychological well-being to a higher extent than it does for other professionals might be a reflection of the increasingly commercial nature of the audit firms [31] that attracts individuals with increasing extrinsic motivation [24] and who thrive in such environment. Regardless, the fact that they are given on average a lower salary compared to other business professionals, auditors’ psychological well-being as well as willingness to stay in auditing are at risk. 

## 5. Practical Implications

The findings of our study suggest that should the auditing industry want to compete with other industries and counter its high staff turnover, it should consider its pay structure and reward system. Increasing salaries might cause the auditing industry to be more competitive while simultaneously increasing the well-being of audit firm employees and further increasing their job satisfaction. However, the audit firm structure and reward system is generally based on the ideas of tournament theory [35], which implies that employees are rewarded according to their rank in the organization, rather than according to their current contribution to the business. Hence, greater compensation is given to partners, who have already put in enough effort previously in their career to gain their current position [35]. A change in the reward system of audit firms would thus be at the current partners’ expense. While this might be a discouraging idea for the current partners, this change might be necessary and timely in order to both increase audit firms employees’ psychological well-being but also to prevent increasing turnover of audit firm employees.

## 6. Limitations and Future Research

The study has a number of limitations that could yet present opportunities for future research. Firstly, we cannot discount the risks of the respondent bias, where those people that felt psychologically well chose to answer the survey, while those with lower levels of psychological well-being have opted out. One way forward in addressing this limitation of the study would be to perform studies employing qualitative design with a focus on the experiences of psychological well-being of individuals that left the business profession or/and audit profession. Secondly, and given the cross-sectional design of the study, we could not explore the causality. Future research could adopt longitudinal or experimental designs to explore the causal links between the variables. Thirdly, the empirically and theoretically motivated focus on auditors vis-à-vis the other business professionals did not allow for comparison of other industry/hierarchy specific groups of professionals. Yet this provides further opportunities for research where for example public servants or executives with business background could be compared with the other business professionals. Furthermore, the study relies on the multivariate linear analysis, motivated by the nature of the variables, which influences the presentation and interpretation of the results, but we acknowledge that other statistical techniques such as logistic or ordinal regression analyses could have further nuanced the paper’s findings. The fifth limitation stems from the data collection method which made it impossible to perform a proper and systematic analysis of the data loss, and where we were limited in the opportunity to influence the independent variables and their operationalization. This means that we cannot claim generalizability of our results to members of the audit industry, Swedish trade union of business professionals’ members, nor to the population of business professionals at large, which makes this study exploratory in nature. However, the method of data collection did provide a unique opportunity to collect a large amount of data from different industries. 

## 7. Conclusions 

The findings of this research contribute to two fields of research. On the one hand, our study contributes to the field of occupational health by uncovering how demography, work conditions as well as job and life satisfaction are associated with the psychological well-being of business professionals. On the other hand, this study contributes to the field of auditing by suggesting how differences between auditors and other business professionals might represent potential reasons for high turnover in the industry. Drawing on the tournament theory [35], our study contributes to both fields by suggesting how industry-specific structures in terms of reward and career advancements could provide an explanation for the difference, between sub-groups of business professionals, regarding triggers of psychological well-being. Finally, and to our knowledge, this is the first study that does not only address the auditors’ psychological well-being as an isolated phenomenon but puts the well-being of that group into a wider business professional context.

## Figures and Tables

**Table 1 ijerph-17-03346-t001:** Descriptive statistics and audit industry differences.

Variables	Total Sample (*n* = 6676)			Audit Industry Excluded (*n* = 6228)			Audit Industry (*n* = 448)			*p*-Value Industry Difference
	Mean	S.D.	% (n)	Mean	S.D.	% (n)	Mean	S.D.	% (n)	
GHQ-12	9.66	4.910		9.63	4.916		10.43	5.281		0.001
Age	42.7	10.073		43.23	9.904		35.53	9.657		0.000
Gender										0.021
Male (0)			44.8 (2991)			45.1 (2809)			39.5 (177)	
Female (1)			55.2 (3685)			54.9 (3419)			60.6 (271)	
EmpChange										0.000
No (0)			79.8 (5327)			80.4 (5007)			71.9 (322)	
Yes (1)			20.2 (1349)			19.6 (1221)			28.1 (126)	
OverComp										0.000
No (0)			53.8 (3592)			56.6 (3525)			14.5 (65)	
Yes (1)			46.2 (3084)			43.4 (2703)			85.5 (383)	
Overtime	1.45	0.995		1.43	0.995		1.74	0.944		0.000
Salary	55.18	26.768		56.31	27.050		39.53	15.616		0.000
Manager										0.000
No (0)			69.5 (4640)			68.2 (4247)			87.7 (393)	
Yes (1)			30.5 (2036)			31.8 (1981)			12.3 (55)	
JobSatisf.	4.97	1.404		4.96	1.407		5.08	1.350		0.085
LifeSatisf.	5.56	1.129		5.57	1.122		5.42	1.121		0.009
Industry										
Audit			6.7 (448)							
Service			22.9 (1527)							
Staffing			1.9 (129)							
IT			8.2 (549)							
Telecom			3.1 (210)							
Bank			10.9 (729)							
Insurance			2.6 (176)							
Trade			9.7 (646)							
Manufacturing			15.8 (1053)							
Non-profit			1.5 (100)							
Other			16.6 (1109)							

Abbreviations: GHQ-12 (12 item General Health Questionnaire); EmpChange (Employer Change); OverComp (Overtime compensation); JobSatisf (Job satisfaction); LifeSatisf (Life satisfaction).

**Table 2 ijerph-17-03346-t002:** Pearson correlation matrix (*n* = 6676).

	Variables	1	2	3	4	5	6	7	8	9	10	11a	11b	11c	11d	11f	11f	11g	11h	11i	11j
1	GHQ-12																				
2	Age	−0.056 **																			
3	Gender	0.111 **	−0.075 **																		
4	EmpChange	−0.034 **	−0.224 **	0.012																	
5	OverComp	−0.016	−0.177 **	0.090 **	0.061 **																
6	Overtime	0.081 **	0.049 **	−0.097 **	−0.035 **	−0.121 **															
7	Salary	−0.130 **	0.396 **	−0.214 **	−0.141 **	−0.321 **	0.291 **														
8	Manager	−0.114 **	0.219 **	−0.100 **	−0.087 **	−0.196 **	0.269 **	0.443 **													
9	JobSatisf.	−0.479 **	0.002	−0.026 *	0.099 **	0.037 **	0.017	0.115 **	0.108 **												
10	LifeSatisf.	−0.544 **	0.021	0.000	0.001	0.008	−0.066 **	0.072 **	0.069 **	0.367 **											
11	a. Audit	0.042 **	−0.192 **	0.028 *	0.053 **	0.211 **	0.078 **	−0.157 **	−0.106 **	0.021	−0.034 **										
11	b. Service	−0.014	0.151 **	−0.051 **	−0.090 **	−0.074 **	0.045 **	0.078 **	0.69 **	−0.005	0.014	−0.146 **									
11	c. Staffing	0.010	−0.053 **	0.032 **	0.155 **	0.103 **	−0.069 **	−0.085 **	−0.074 **	−0.037 **	−0.035 **	−0.038 **	−0.076 **								
11	d. IT	−0.009	0.017	0.038 **	0.026 *	−0.046 **	−0.029 *	−0.042 **	0.001	0.016	0.005	−0.080 **	−0.163 **	−0.042 **							
11	e. Telecom	−0.001	0.045 **	0.003	−0.035 **	−0.059 **	−0.027 *	0.044 **	−0.026 *	−0.032 **	0.006	−0.048 **	−0.098 **	−0.025 *	−0.054 **						
11	f. Bank	0.003	−0.024 *	−0.025 *	−0.057 **	0.035 **	−0.045 **	0.015	−0.089 **	0.007	0.014	−0.094 **	−0.191 **	−0.049 **	−0.105 **	−0.063 **					
11	g. Insurance	−0.011	0.015	−0.021	−0.013	−0.040 **	−0.041 **	0.040 **	−0.032 **	−0.001	0.004	−0.044 **	−0.090 **	−0.023	−0.049 **	−0.030 *	−0.058 ’’				
11	h. Trade	−0.011	−0.036**	0.014	0.005	−.053 **	0.024 *	0.010	0.080 **	−0.013	0.018	−0.088 **	−0.178 **	−0.046 **	−0.098 **	−0.059 **	−0.115 **	−0.054 **			
11	i. Manufacturing	0.008	−0.008	0.014	0.030 *	−0.012	0.009	−0.032 **	0.025*	−0.006	−0.003	−0.116 **	−0.236 **	−0.061 **	−0.130 **	−0.078 **	−0.152 **	−0.071 **	−0.142 **		
11	j. Non-profit	0.009	0.034 **	0.012	−0.004	0.029 *	−0.008	−0.037 **	0.034 **	0.013	−0.013	−0.033 **	−0.067 **	−0.017	−0.037 **	−0.022	−0.043 **	−0.020	−.040 **	−0.053 **	
11	k. Other	−0.004	−0.017	0.053 **	0.026 *	−0.026 *	−0.014	−0.003	0.021	0.014	−0.006	−0.120 **	−0.243 **	−0.063 **	−0.134 **	−0.080 **	−0.156 **	−0.073 **	−0.146 **	−0.193 **	−0.055 **

* Correlation is significant at the 0.05 level; ** Correlation is significant at the 0.01 level. Abbreviations: GHQ-12 (12 item General Health Questionnaire); EmpChange (Employer Change); OverComp (Overtime compensation); JobSatisf (Job satisfaction); LifeSatisf (Life satisfaction).

**Table 3 ijerph-17-03346-t003:** Regression models (Dependent variable: GHQ-12).

	Model 1		Model 2		Model 3	
Variables	(*n* = 6676)		(*n* = 6228)		(*n* = 448)	
	β	S.E.	β	S.E.	β	S.E.
Age	−0.010 *	0.005	−0.009	0.005	−0.021	0.023
Gender	0.972 **	0.096	1.007 **	0.099	0.197	0.384
EmpChange	−0.160	0.122	−0.158	0.125	−0.935 *	0.454
OverComp	−0.292 **	0.101	−0.284 **	0.102	−0.663	0.570
Overtime	0.453 **	0.051	0.482 **	0.052	0.063	0.207
Salary	−0.010 **	0.002	−0.009 **	0.002	−0.043 **	0.016
Manager	−0.475 **	0.116	−0.515 **	0.117	0.102	0.595
JobSatisf.	−1.103 **	0.036	−1.090 **	0.037	−1.208 **	0.158
LifeSatisf.	−1.814 **	0.045	−1.805 **	0.046	−1.912 **	0.179
Industry						
Audit	0.280	0.215				
Service						
Staffing	−0.581	0.357				
IT	0.110	0.190				
Telecom	−0.172	0.280				
Bank	0.125	0.173				
Insurance	−0.106	0.303				
Trade	−0.119	0.179				
Manufacturing	0.036	0.153				
Non-profit	0.348	0.393				
Other	−0.044	0.151				
Constant	25.309 **		25.044 **		29.963 **	
F-value	243.072 **		471.750 **		40.898 **	
Adj R2	40.8%		40.5%		44.5%	
Higest VIF	1.653		1.584		1.889	

* *p* < 0.05; ** *p* < 0.01. Specification of dichotomous variables: Gender (female); EmpChange (changed employer since October 2016) OverComp (overtime compensation in the form of money and/or time); Manager (operational, budget and/or personnel manager). Specification of reference group: Industry (Service).

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
