# Peer review of "Antecedents of Psychological Well-Being among Swedish Audit Firm Employees"

_ijerph, 2020, doi:10.3390/ijerph17103346_

Round 1

Reviewer 1 Report

Dear Authors I carefully evaluated the study, finding it overall well written. The theme is interesting and there is a need to investigate these aspects. Nevertheless, some main concern are present and they need to be solved before considering the paper suitable for publication. Introduction: a better explanation of the “turnover due to the ‘up-or-out’ system” is needed. You stated “All of these reasons for employee turnover are likely to be strengthened in times of economic boom, leading to staff shortages”. This sentence is questionable. This is not only a problem in time of economic boom, considering that we are not in an economic boom period. I suggest also to consider how a period of economic crisis can affect organizational process and consequently the workload of the employees. “A shortage of personnel increases the risk of long term stress and illness”: reference is needed “A personnel shortage affects workers’ ability to maintain a satisfactory level of performance”: reference is needed “In turn, this adversely affects job satisfaction in particular, and psychological well-being in general”: reference is needed “The paper first presents a literature review; this is followed by the method and a discussion of the results and analysis. The paper concludes with a discussion that includes both theoretical and practical contributions”. You must follow the rules of the journal. It is not necessary to describe how you structured the paper. The Theory section have to be included in the introduction. It must result in a unique paragraph “introduction” where you “Explain the scientific background and rationale for the investigation being reported” and “State specific objectives, including any prespecified hypotheses” (as reported in the STROBE guidelines for observational studies). Splitting the scientific background in two parts worsens the readability of the introduction. Methods: GHQ-12: state clearly what method of interpretation you used in your study. Limit the presentation about your study, excluding from description all the methods not pertinent with your study. How did you evaluate monthly salary? Add a specific section about statistical analysis at the end of the methods section. I suggest to add some subheadings along the methods section (e.g. study design, study population, procedures, measurements and statistical analysis. They could allow the reader the find out easily basic information about the study conduction. Results: move from results to discussion all references and comparisons with other studies. Comparisons must be done in discussion section. No references are needed in the result section. You only must present your results, without any comparison. Table 1. What kind of test did you perform to compare the two subgroups? All the statistical methods have to be reported in the methods section. Add frequencies (N) along with the percentage. Table 2. You used a Pearson matrix correlation. Have you tested for normal distribution of the variables? Indicate dependent and independent variables. Explain all the abbreviations. You stated: “It is important to note that the correlation matrix does not indicate any multicollinearity problems, as none of the pairwise correlations are close to the critical value of 0.8.” Support this statement with some relevant references. Moreover, I noticed that some r values are about 0.8 (e.g 11a vs 11h; 11g vs 11b). A collinearity concern could be argued. Multivariate analysis: add the reference group for all the independent variables (e.g. gender, job task and so on). Replace the term correlation with association when you talk about regression results. Discussion is too brief. Following STROBE guidelines for observational studies, in the discussion section authors must • Summarise key results with reference to study objectives • Discuss limitations of the study, taking into account sources of potential bias or imprecision. Discuss both direction and magnitude of any potential bias • Give a cautious overall interpretation of results considering objectives, limitations, multiplicity of analyses, results from similar studies, and other relevant evidence • Discuss the generalisability (external validity) of the study results All these elements are lacking in your discussion. Conclusions are also lacking. State clearly what this paper adds to the literature about the theme and how your results can contribute to bridge a literature gap.

Author Response

Dear Reviewer 1

We acknowledge and appreciate the positive and constructive spirit in which the reviews are made. We hope that you feel that we share that attitude. Thank you very much for your effort and interest in our research!

You can find your numbered comments below followed by numbered responses to your comments with specification of changes made.

Q1: Introduction: a better explanation of the “turnover due to the ‘up-or-out’ system” is needed.

R1: Thank you for pointing out the need for clarification. We have now added an explanation supported by two references.

Q2: You stated “All of these reasons for employee turnover are likely to be strengthened in times of economic boom, leading to staff shortages”. This sentence is questionable. This is not only a problem in time of economic boom, considering that we are not in an economic boom period. I suggest also to consider how a period of economic crisis can affect organizational process and consequently the workload of the employees.

R2: Thank you for pointing it out and the opportunity to clarify our statement. Apart from the fact that inflation has not picked up since the economic crisis in 2008, other indicators point to the fact that we have been in an economic boom that peaked in 2018. Not at least regarding job availability. According to the Swedish Association of Local government and Regions (SALAR) and Statistics Sweden (SCB); potential job hours in society have exceeded actual job hours during the period 2009 – 2017 (while actual job hours continuously have increased during the same period). However, since December 2017 SALAR has in their forecasts assumed that the economic boom will culminate in 2018 (E.g. SALAR https://webbutik.skr.se/bilder/artiklar/pdf/7585-549-3.pdf?issuusl=ignore). However, the key issue here is not the strength in the economy per se, but rather the availability and demand for jobs. Thus, we have rephrased the text also included a reference. By pointing out the moderating effect, we also consider times of economic crises and low job demand/job availability.

 Q3: “A shortage of personnel increases the risk of long term stress and illness”: reference is needed “A personnel shortage affects workers’ ability to maintain a satisfactory level of performance”: reference is needed “In turn, this adversely affects job satisfaction in particular, and psychological well-being in general”: reference is needed “

R3: To strengthen our statements we have now added several references and slightly rephrased the statements/sentences you have pointed out. These changes can now be found on pg. 2.

Q4: “The paper first presents a literature review; this is followed by the method and a discussion of the results and analysis. The paper concludes with a discussion that includes both theoretical and practical contributions”. You must follow the rules of the journal. It is not necessary to describe how you structured the paper. The Theory section have to be included in the introduction. It must result in a unique paragraph “introduction” where you “Explain the scientific background and rationale for the investigation being reported” and “State specific objectives, including any prespecified hypotheses” (as reported in the STROBE guidelines for observational studies). Splitting the scientific background in two parts worsens the readability of the introduction.

R4: Thank you for this comment and apologies for not following the guidelines. We have now adjusted the paper – the section describing how we structured the paper is removed and the previous introduction and theory sections are now merged into one introduction section.

 Q5: Methods: GHQ-12: state clearly what method of interpretation you used in your study. Limit the presentation about your study, excluding from description all the methods not pertinent with your study. How did you evaluate monthly salary? Add a specific section about statistical analysis at the end of the methods section. I suggest to add some subheadings along the methods section (e.g. study design, study population, procedures, measurements and statistical analysis. They could allow the reader the find out easily basic information about the study conduction.

R5: Thank you for pointing out these issues. We agree that there was a need for clarification. We have now clearly stated the method of interpretation regarding GHQ-12, removed most text about methods not pertinent with our study, clarified how we evaluated monthly salary and added a specific section about statistical analyses. We have also added subheadings throughout the method section. In addition to the issues pointed out in your comment we also added a clarification regarding the design of the study.

Q6: Results: move from results to discussion all references and comparisons with other studies. Comparisons must be done in discussion section. No references are needed in the result section. You only must present your results, without any comparison.

R6: We have now removed these comparisons and references.

Q7: Table 1. What kind of test did you perform to compare the two subgroups? All the statistical methods have to be reported in the methods section. Add frequencies (N) along with the percentage.

R7: We have now added a specific section about statistical analysis in the method chapter (which includes specification of independent sample t-tests and Pearson chi-square tests as the methods used to compare the two subgroups) and we have added frequencies in Table 1.

Q8: Table 2. You used a Pearson matrix correlation. Have you tested for normal distribution of the variables? Indicate dependent and independent variables. Explain all the abbreviations.

R8: In this study employ the normality assumption (as described and explained by. e.g. Gujarati, D.N. Basic Econometrics. McGraw Hill 2003, fourth international edition.). According to the central limit theorem (CLT) of statistics the distribution of the sum of a large number of independent random variables is approximately normally distributed (even if the original variables themselves are not normally distributed) and thus, CLT provides a theoretical justification for assuming normality. I.e. due to the large sample in this study, we assume the dependent and independent variables to be approximately normaly distributed

In line with your suggestion, we have added explanations for all the abbreviations used in Table 2 (under the table).

Q9: You stated: “It is important to note that the correlation matrix does not indicate any multicollinearity problems, as none of the pairwise correlations are close to the critical value of 0.8.” Support this statement with some relevant references. Moreover, I noticed that some r values are about 0.8 (e.g 11a vs 11h; 11g vs 11b). A collinearity concern could be argued.

R9Thank you for pointing this out. We have now added a reference to a textbook in basic econometrics, to add the credibility to our statement. Maybe our correlation matrix was previously too small and it was not clear what the correlation coefficients were.

The correlation between 11a vs 11h is 0.088 which is < 0.8.

The correlation between 11b vs 11g is 0.09 which is < 0.8.

Thus, the correlation matrix does not indicate any collinearity problems at all. Neither does the additional VIF test reported in table 3.

Q10: Multivariate analysis: add the reference group for all the independent variables (e.g. gender, job task and so on). Replace the term correlation with association when you talk about regression results.

R10: We have now added specifications of dichotomous variables and of the reference group for the industry variable under table 3. We have also replaced the term correlation with association when talking about the regression results.

 Q11: Discussion is too brief. Following STROBE guidelines for observational studies, in the discussion section authors must

  • Summarize key results with reference to study objectives
  • Discuss limitations of the study, taking into account sources of potential bias or imprecision.
  • Discuss both direction and magnitude of any potential bias
  • Give a cautious overall interpretation of results considering objectives, limitations, multiplicity of analyses, results from similar studies, and other relevant evidence
  • Discuss the generalizability (external validity) of the study results

All these elements are lacking in your discussion.

R11: Thank you for this comment and apologies for overly short discussion and not following the STORBE guidelines. Guided by these guidelines and your suggestions we have now expanded in our discussion, which included a neutral summary of the key results in relationship to study objective, followed by interpretation of the results with the help of the past research. After the discussion session we have now added practical implication section, followed by the extensive section presenting the limitations of our study that brings up the sources of potential biases, their magnitude and directions as well as discusses the generalizability of the study results. We also intertwine future research directions into this discussion. These sections can be found on pg. 11 – 15.

Q12: Conclusions are also lacking. State clearly what this paper adds to the literature about the theme and how your results can contribute to bridge a literature gap.

R12: Thank you for pointing it out, we have now followed your suggestion and added the conclusion section to our study where we present the contribution of our research and identify the literature gap that our study is filling. This section can be found on pg. 15.

Reviewer 2 Report

Dear authors,

I have reviewed your manuscript and I appreciate the effort you have done on it. Furthermore, I find it interesting how you have included variables related to well-being in this kind of studies. I also appropriate that the authors show some of the research limitations, which I had identified myself, so it will not be part of my comments. However, I consider that the manuscript needs to improve some issues  to be published:

Related to the introduction, there is no theoretical reference model or models (for example, that justify the use of GHQ and SWL separately) and that support the findings, or previous literature that have explored the relationships between the different variables studied.
Regarding the method, I consider that it is necessary to incorporate additional information and eliminate some other. Limitations are not usually part of the method, because they are not just about it. The limitations have to be incorporated in the final part of the discussion, in addition to being extended. It is necessary to incorporate more information in relation to the ethical conditions of the work, or more detailed aspects such as how long the work was applied (not just autumn), more detail of the characteristics of the sample, if there were other reasons for exclusion or not, etc.
The results summarize the progress of the work. However, I wonder if, despite the sample being very wide, the difference between subsamples would not imply having to carry out logistical analyzes for not meeting the assumptions. Has this been verified? It would be reasoneable to give information about it.
Furthermore, if there are many differences between the subsamples, how can we attribute the wellbeing results to being in a subgroup and not due to some or some of its characteristics? Argue or include in limitations.
Finally, the discussion needs to be improved, in more depth, which may be able to be formulated when the introduction is strengthened. It is necessary to make more clear what this work really contributes. In addition to incorporating limitations, incorporating future lines of work and practical implications.

Hopefully you can improve your manuscript.

All the best.

Author Response

Dear Reviewer 2

We acknowledge and appreciate the positive and constructive spirit in which the reviews are made. We hope that you feel that we share that attitude. Thank you very much for your effort and interest in our research!

You can find your numbered comments below followed by numbered responses to your comments with specification of changes made.

Q1: There is no theoretical reference model or models (for example, that justify the use of GHQ and SWL separately) and that support the findings, or previous literature that have explored the relationships between the different variables studied.

R1: Thank you for bringing this up, we apologize for not being specifically clear on which specific literature we draw when we explore GHQ and SWL separately, as well the variables the are associate with it.  In the text we have now made it explicit that we followed seminal study by Dana & Griffin (1999) that in their conceptual/review paper dealing with health/and well-being in the workplace separate psychological aspects of health and well-being from that of  job and life satisfaction well-being dimensions and who further identify different triggers of psychological well being that we now employ in our paper. To make this clear we now mention it in the Introduction as well as in the Method sections of our paper.

Q2: Regarding the method, I consider that it is necessary to incorporate additional information and eliminate some other. Limitations are not usually part of the method, because they are not just about it. The limitations have to be incorporated in the final part of the discussion, in addition to being extended.

R2: We agree with the reviewer that putting limitation in the method part was a mistake, we have now removed this discussion. In line with your and other reviewers comments we have now added an extensive section called Limitations and Future Research directions, that discuss study’s limitations, brings up the sources of potential biases, their magnitude and directions, discusses the generalizability of the study results as well as suggest the future research directions. This section can be found on pg. 14-15.

Q3: It is necessary to incorporate more information in relation to the ethical conditions of the work, or more detailed aspects such as how long the work was applied (not just autumn, more detail of the characteristics of the sample, if there were other reasons for exclusion or not, etc.

R3: Thank you for this comment and the opportunity to clarify some aspects of our study. We have now clarified the time period for data collection (October to December 2017), added more information regarding the work process as well as more details regarding the sample.

Q4: The results summarize the progress of the work. However, I wonder if, despite the sample being very wide, the difference between subsamples would not imply having to carry out logistical analyzes for not meeting the assumptions. Has this been verified? It would be reasonable to give information about it. Furthermore, if there are many differences between the subsamples, how can we attribute the wellbeing results to being in a subgroup and not due to some or some of its characteristics? Argue or include in limitations.

R4: Thank you for this insightful suggestion. We have carried out several different alternative regressions to the test robustness of our models and results. We have carried out the analyzes using different combinations of the independent variables, especially we have focused on the industry variables and we have e.g. run regressions using just only the audit industry variable (i.e. audit industry 1, all other industries 0). The result does not differ in these different models and thus, we suggest the results are robust. We have also run regressions excluding each independent variable separately to make sure that the well-being results are not explained by other subgroups or characteristics.     

We have now also clarified this and described out robustness tests in the paper.

When it comes to the performance of logistic regression, if we understand the reviewer, suggestion was to dichotomize the dependent variable to test our findings with probit. That could have been an option, but potential variation of the data would have been lost with dichotomization as well as certain theoretical assumptions in relation to psychological well-being. Yet we think it could be a very good idea for future research. Inspired by this comment we now took the liberty to include it into the Limitations and Future research directions section of the paper.

Q5: Finally, the discussion needs to be improved, in more depth, which may be able to be formulated when the introduction is strengthened. It is necessary to make more clear what this work really contributes. In addition to incorporating limitations, incorporating future lines of work and practical implications.

R5: Thank you for this comment that resonated well with the comments from the other reviewers.  Guided by these your and other reviewer suggestions we have now expanded our discussion, which now includes a neutral summary of the key results in relationship to study objective, followed by interpretation of the results with the help of the past research. After the discussion session we have now added practical implication section, followed by the extensive section presenting the limitations and future research directions. The paper now concludes with overarching conclusions section where we outline our study’s contributions. These changes can be found on pg. 11 – 15.

Reviewer 3 Report

I read this paper with great interest. It is very well-organized, and literatures are sufficiently reviewed. Their research focus, i.e. the audit industry, is a very interesting topic as it is often characterized by high-work load and physical health concerns of employees. It is therefore quite surprising to me that the results show a higher level of job satisfaction among employees in the auditing industry than other industries on average, though their life satisfaction is slightly lower.

Based on the results that “salary” has a significant and negative correlation to GHQ-12, the author reasonably suggested, in the discussion part, that the salary and reward system in the auditing industry should be changed.

It is also very interesting that the variable “employer change” is significantly related to the well-being only in the auditing industry. Perhaps, for people working in the auditing industry, the working relationships between employers and employees and a new job environment/company culture may present some positive influences on employees’ well-being. Do the author have any plans to further their studies related to this (and/or other variables)? It would be nice to hear from them their future research directions.

Author Response

Dear Reviewer 3

We acknowledge and appreciate the positive and constructive spirit in which the reviews are made. We hope that you feel that we share that attitude. Thank you very much for your effort and interest in our research!

Q1: I read this paper with great interest. It is very well-organized, and literatures are sufficiently reviewed. Their research focus, i.e. the audit industry, is a very interesting topic as it is often characterized by high-work load and physical health concerns of employees. It is therefore quite surprising to me that the results show a higher level of job satisfaction among employees in the auditing industry than other industries on average, though their life satisfaction is slightly lower.

Based on the results that “salary” has a significant and negative correlation to GHQ-12, the author reasonably suggested, in the discussion part, that the salary and reward system in the auditing industry should be changed.

It is also very interesting that the variable “employer change” is significantly related to the well-being only in the auditing industry. Perhaps, for people working in the auditing industry, the working relationships between employers and employees and a new job environment/company culture may present some positive influences on employees’ well-being. Do the author have any plans to further their studies related to this (and/or other variables)? It would be nice to hear from them their future research directions.

R1. Thank you for these insightful comments and summary of our paper. As reviewer will see we have now incorporated some of these suggestions into our extended discussion section as well as added additional section of Limitations and Future Research suggestions where we expand on future steps that can be taken in this field of research

Reviewer 4 Report

The study is somewhat exploratory. The survey resulted in approximately 11 000 respondents in total, of which 6 676 were useful for this study. (That is, these respondents answered the questions regarding well-being and were working when the survey was distributed in October 2017, while respondents who were, for example, unemployed during this time period were excluded.) A limitation resulting from this data collection method is that it is impossible to perform a proper and systematic analysis of the data loss, and there was limited opportunity to influence the independent variables and their operationalization. However, this method provided a unique opportunity to collect a large amount of data from different  industries. The methods are described in detail.

To measure the dependent variable – namely, psychological well-being – the General Health Questionnaire (GHQ-12) was used. GHQ-12 is a widely used unidimensional self-assessment  instrument that focuses on the last two weeks and uses 12 questions to measure psychological health problems including the dimensions of anxiety and depression, social dysfunction and loss of confidence.

The results show that the respondents - generally young people, appear to be satisfied with their jobs and their lives. The comparison between the two subgroups indicates that the respondents from the auditindustry differ in many aspects from the other respondents in the sample. The respondents from the audit industry show a higher GHQ-12 score and differ in the other variables. A greater proportion of the respondents from the audit industry are female; moreover, the respondents from the audit industry are younger than other respondents in the total sample. Even though a greater proportion  of the audit subgroup receives overtime compensation in comparison with the other subgroup, the  auditors earn less on average and seem to change employers more often.

Conclusion. The introduction provides sufficient background and contains all relevant references. The research plan is properly prepared and implemented. The methods have been properly described. The results at work are clearly presented. Discussion carried out correctly.

Author Response

Dear Reviewer 4

We acknowledge and appreciate the positive and constructive spirit in which the reviews are made. We hope that you feel that we share that attitude. Thank you very much for your effort and interest in our research!

Q1: The study is somewhat exploratory. The survey resulted in approximately 11 000 respondents in total, of which 6 676 were useful for this study. (That is, these respondents answered the questions regarding well-being and were working when the survey was distributed in October 2017, while respondents who were, for example, unemployed during this time period were excluded.) A limitation resulting from this data collection method is that it is impossible to perform a proper and systematic analysis of the data loss, and there was limited opportunity to influence the independent variables and their operationalization. However, this method provided a unique opportunity to collect a large amount of data from different  industries. The methods are described in detail.

To measure the dependent variable – namely, psychological well-being – the General Health Questionnaire (GHQ-12) was used. GHQ-12 is a widely used unidimensional self-assessment  instrument that focuses on the last two weeks and uses 12 questions to measure psychological health problems including the dimensions of anxiety and depression, social dysfunction and loss of confidence.

The results show that the respondents - generally young people, appear to be satisfied with their jobs and their lives. The comparison between the two subgroups indicates that the respondents from the auditindustry differ in many aspects from the other respondents in the sample. The respondents from the audit industry show a higher GHQ-12 score and differ in the other variables. A greater proportion of the respondents from the audit industry are female; moreover, the respondents from the audit industry are younger than other respondents in the total sample. Even though a greater proportion  of the audit subgroup receives overtime compensation in comparison with the other subgroup, the  auditors earn less on average and seem to change employers more often.

Conclusion. The introduction provides sufficient background and contains all relevant references. The research plan is properly prepared and implemented. The methods have been properly described. The results at work are clearly presented. Discussion carried out correctly.

R1: Thank you for these insightful comments and summary of our paper!

Round 2

Reviewer 1 Report

Dear Authors

I find the paper substantially improved.

Well done.

Best regards

Reviewer 2 Report

I appreciate the effort to improve the manuscript. Good work!